# Single Administration of the T-Type Calcium Channel Enhancer SAK3 Reduces Oxidative Stress and Improves Cognition in Olfactory Bulbectomized Mice

**DOI:** 10.3390/ijms22020741

**Published:** 2021-01-13

**Authors:** Dian Yuan, An Cheng, Ichiro Kawahata, Hisanao Izumi, Jing Xu, Kohji Fukunaga

**Affiliations:** Department of Pharmacology, Graduate School of Pharmaceutical Sciences, Tohoku University, Sendai 980-8578, Japan; yuan.dian.t6@dc.tohoku.ac.jp (D.Y.); cheng.an.q6@dc.tohoku.ac.jp (A.C.); kawahata@tohoku.ac.jp (I.K.); fukunaga@mail.pharm.tohoku.ac.jp (H.I.); xu.jing.q7@dc.tohoku.ac.jp (J.X.)

**Keywords:** Alzheimer’s disease, oxidative stress, T-type calcium channel, SAK3, microglia

## Abstract

Alzheimer’s disease (AD), characterized by cognitive impairments, is considered to be one of the most widespread chronic neurodegenerative diseases worldwide. We recently introduced a novel therapeutic agent for AD treatment, the T-type calcium channel enhancer ethyl-8-methyl-2,4-dioxo-2-(piperidin-1-yl)-2H-spiro[cyclopentane-1,3-imidazo[1,2-a]pyridin]-2-ene-3-carboxylate (SAK3). SAK3 enhances calcium/calmodulin-dependent protein kinase II and proteasome activity, thereby promoting amyloid beta degradation in mice with AD. However, the antioxidative effects of SAK3 remain unclear. We investigated the antioxidative effects of SAK3 in olfactory bulbectomized mice (OBX mice), compared with the effects of donepezil as a positive control. As previously reported, single oral administration of both SAK3 (0.5 mg/kg, p.o.) and donepezil (1.0 mg/kg, p.o.) significantly improved cognitive and depressive behaviors in OBX mice. Single oral SAK3 administration markedly reduced 4-hydroxy-2-nonenal and nitrotyrosine protein levels in the hippocampus of OBX mice, which persisted until 1 week after administration. These effects are similar to those observed with donepezil therapy. Increased protein levels of oxidative stress markers were observed in the microglial cells, which were significantly rescued by SAK3 and donepezil. SAK3 could ameliorate oxidative stress in OBX mice, like donepezil, suggesting that the antioxidative effects of SAK3 and donepezil are among the neuroprotective mechanisms in AD pathogenesis.

## 1. Introduction

Alzheimer’s disease (AD) is one of the most common progressive neurodegenerative disorders worldwide and is also the most common form of dementia. AD is mainly characterized by the presence of β-amyloid (Aβ) and intracellular neurofibrillary tangles in the brain [1,2,3]. Over the past few years, a large number of studies have indicated that oxidative stress is closely associated with the etiology and pathology of neuronal degeneration and cell death in the brain of patients with AD [4]. More specifically, in early AD, which is characterized by mild cognitive impairment, protein oxidation is significantly increased in the hippocampus prior to the deposition of Aβ [5]. Additionally, previous investigations have reported that donepezil plays a key role in restoring redox homeostasis by inhibiting the activation of the enzyme, acetylcholinesterase, thus displaying antioxidative effects in patients with AD [6]. Furthermore, our previous research revealed that oral administration of glutathione, a potent antioxidant drug, ameliorated elevated oxidative stress in the brains of AD mice and rescued cognitive impairments observed in behavioral tests [7]. Moreover, we concluded that antioxidant drugs play a vital role in AD therapy. To facilitate research on AD pathogenesis, many animal models have been developed, including injection models, olfactory bulbectomy models and rapidly aging rodent models. Among these, olfactory bulbectomized mice (OBX mice) display some symptoms that are similar to human AD, including spatial memory deficits, a significant increase in the level of β-amyloid in the brain and cholinergic deficits in forebrain neurons [8,9]. In addition, some studies have demonstrated that OBX mice exhibit depression-like symptoms in behavioral tasks, as well as cellular hallmarks of depression; therefore, these mice are widely used in the testing of antidepressant drugs [10]. Increased levels of oxidative damage in the brain have also been observed in the OBX mouse model and these mice were very sensitive to treatment with antioxidant therapeutic drugs [11,12]. Moreover, because of its comparatively easy and rapid reproducibility of AD characteristics, the OBX mice is suitable for pathological research of AD [13].

Oxidative stress, which is caused by an imbalance between reactive oxygen and nitrogen species and antioxidant defenses [14], has been defined as a toxic factor to some biological molecules, such as lipids, proteins and DNA [15]. Furthermore, many reports have revealed that the level of reactive oxygen species markers, including lipid peroxidation [16], DNA/RNA [17] and protein oxidation [18] increase considerably in each part of the brains of AD subjects. However, the specific relationship between AD pathogenesis and oxidative stress remains unclear and needs to be fully elucidated. We hypothesized that oxidative stress plays a key role in AD pathogenesis and it is therefore meaningful to perform further experiments to clarify their interaction. In this study, we used 4-hydroxynonenal (4-HNE), a major secondary product of lipid peroxidation and nitrotyrosine, a product derived from nitric oxide activity [19], to investigate oxidative stress levels in OBX mice.

Previous investigations reported a mutual interaction between neuroinflammation and AD pathogenesis, accompanied by damage to the neurons and neurites in the brain [20]. Furthermore, high expression levels of inflammatory markers, including interleukin-6 (IL-6), were observed in AD patients or in animal models [21,22]. Moreover, for the progression of inflammation in the AD brain, activation of central nervous system glial cells, such as microglia, is very prominent [23]. Microglia, one type of immune cells in the brain, have a resemblance role in neuronal functions [24]. To be more specific, microglial masses have appeared in the aggregated Aβ deposit sites in the AD brain, along with the production of some proinflammatory cytokines, which contributed to the progression of AD [25,26]. In contrast, some hypotheses suggest that microglia exhibit neuroprotective effects in AD due to their promotion of Aβ deposition clearance and their ability to discard glutamate [27,28]. Given that microglia are prominent both in the inflammatory response and AD pathogenesis, it is meaningful to include them in an intensive study.

T-type calcium channels and low-voltage-activated calcium channels are related to some disease pathologies, such as cancer [29], pain [30] and Parkinson’s disease [31]. Cav3.1, a type of T-type calcium channel that is widespread in the thalamic relay neurons, regulates the response in the auditory cortex [32]. However, Cav3.2 and Cav3.3, which are enriched in gamma aminobutyric acid neurons, are closely associated with neuropathic pain [33]. We recently developed an T-type calcium channel enhancer, SAK3 (ethyl-8-methyl-2,4-dioxo-2-(piperidin-1-yl)-2H-spiro[cyclopentane-1,3-imidazo [1,2-a] pyridin]-2-ene-3-carboxylate). Based on whole cell patch-clamp analysis, SAK3 (0.01–10 nM) significantly enhanced Cav3.1 current in neuro2A cells ectopically expressing Cav3.1. SAK3 (0.1–10 nM) also enhanced Cav3.3 but not Cav3.2 currents in the transfected cells [34]. Besides, our previous results indicated that acute oral administration of SAK3 could increase acetylcholine (ACh) release in the hippocampus and restore cognitive impairment in OBX mice, using the most effective dose of 0.5 mg/kg but not 0.1 mg/kg [34]. Some drugs, such as donepezil and galantamine, have already been used for the clinical treatment of AD. Donepezil, a potent acetylcholinesterase inhibitor, produced a protective effect on cholinergic neurons in OBX mice under the most effective dose of 1.0 mg/kg [35]. Although the ability of SAK3 to restore memory loss in OBX mice has been reported in our previous work, its effects on oxidative damage in the mouse brain remain unclear. We therefore used the OBX mouse model to perform further research in the present study.

Based on the above reports, we used SAK3 in this study to investigate its pharmacodynamic effects on AD treatment. Furthermore, we also explored the relationship between oxidative stress and AD pathogenesis.

## 2. Results

### 2.1. Effects of Acute SAK3 Administration on Spatial Memory, Cognitive Functions and Depressive-Like Behaviors in OBX Mice

To investigate whether acute SAK3 could ameliorate the cognitive deficits that characterize AD pathogenesis, we performed behavioral analyses using OBX mouse model in mice with AD. In the Y-maze task, there was no difference among all experimental groups in the total number of arm entry number (Figure 1A). In contrast, vehicle-treated OBX mice had a reduced percentage of alternation (53.78 ± 2.434, *p* = 0.034) than the vehicle-treated sham mice; this alteration was significantly restored by SAK3 administration (0.5 mg/kg, 67.79 ± 2.977, *p* = 0.0318) or donepezil (1.0 mg/kg, 62.45 ± 6.411, *p* = 0.2161) (Figure 1B). In the novel object recognition task, there was no difference in the percentage of discrimination index across all groups using the same object in the trial session. Nevertheless, vehicle-treated OBX mice failed to discriminate between the familiar and novel objects and acute administration of SAK3 (0.5 mg/kg, p.o.) or donepezil (1.0 mg/kg, p.o.) restored this ability during the test session (vehicle-treated sham: 58.849 ± 1.598, *p* < 0.0001; SAK3-treated sham: 63.916 ± 1.916, *p* < 0.0001; donepezil-treated sham: 57.742 ± 2.044, *p* < 0.0001; vehicle-treated OBX: 50.000 ± 0.678, *p* > 0.999; SAK3-treated OBX: 61.355 ± 2.596, *p* < 0.0001; donepezil-treated OBX: 60.618 ± 2.352, *p* < 0.0001; all comparisons were made with the familiar object condition; Figure 1C). In the step-through passive avoidance task, we observed no changes in the latency times to enter the dark room from the light room among all groups in the trial session. However, vehicle-treated OBX mice (142.2 ± 43.2, *p* = 0.0064 vs. vehicle-treated sham) had reduced latency time than the sham mice, which was rescued by SAK3 administration (0.5 mg/kg, 274.6 ± 25.45, *p* = 0.0062) and donepezil (1.0 mg/kg, 299 ± 1.00, *p* = 0.0014) in the test session (Figure 1D).

The fact that the AD mice exhibit depressive-like behaviors has already been reported previously [36]. To investigate the effects of acute SAK3 administration on depressive-like behaviors in OBX mice, we conducted a tail suspension task (TST) and a forced swimming task (FST). In the TST, immobile time was significantly longer in vehicle-treated mice (191.2 ± 12.26, *p* = 0.0106) than sham mice but no remarkable improvements were observed after acute SAK3 and donepezil administration (Figure 1E). Moreover, no differences were observed in the FST (vehicle-treated OBX: 262.8 ± 5.98, *p* < 0.0001 vs. vehicle-treated sham, Figure 1F).

### 2.2. Elevated 4-HNE and Nitrotyrosine Protein Levels Were Suppressed by SAK3 Treatment in OBX Mice

Many studies have reported that OBX mice provide a good model for investigating oxidative stress and neuroinflammation [37]. Therefore, in this study, to examine the oxidative stress level in OBX mice and assess the effects of acute SAK3 administration on the oxidation-induced damage, we measured the expression of 4-HNE and nitrotyrosine protein, indicators of lipid oxidative damage. In immunoblotting analyses, it was found that, 4-HNE (304.2 ± 82.99, *p* < 0.05) and nitrotyrosine protein (317.5 ± 81.66, *p* < 0.05) levels increased significantly in the hippocampal CA1 region of vehicle-treated OBX mice than in vehicle-treated sham mice (Figure 2A–C). In contrast, acute SAK3 (4-HNE, 84.26 ± 9.887, *p* < 0.05; nitrotyrosine, 84.26 ± 9.887, *p* < 0.05) and donepezil (4-HNE, 139.4 ± 35.2, *p* < 0.05; nitrotyrosine, 151.2 ± 35.53, *p* < 0.05) administration significantly ameliorated the increase in the levels of both proteins in OBX mice than that in vehicle-treated OBX mice (Figure 2B,C).

### 2.3. The Number of 4-HNE-and Nitrotyrosine-Positive Cells Decreased after SAK3 Treatment in OBX Mice

In the immunohistochemical analyses, the number of 4-HNE-positive cells or nitrotyrosine-positive cells was measured in the hippocampal CA1 region. The number of 4-HNE-positive cells was significantly increased in OBX mice (60.88 ± 4.654, *p* = 0.0175) than in vehicle-treated sham mice, which is indicative of an increase in oxidative stress in the brain (Figure 3A,C). However, the increased numbers were suppressed by acute administration of SAK3 (36.67 ± 4.745, *p* = 0.0030) and donepezil (38.71 ± 4.162, *p* = 0.0043) in OBX mice when compared with that in vehicle-treated mice (Figure 3C). The same changes and drug effects were observed with nitrotyrosine protein (vehicle-treated OBX: 76.00 ± 7.439, *p* = 0.0251 vs. vehicle-treated sham; SAK3-treated OBX: 52.67 ± 5.333, *p* = 0.0444 vs. vehicle-treated OBX; donepezil-treated OBX: 54 ± 3.559, *p* = 0.0412 vs. vehicle-treated OBX, Figure 3B,D).

### 2.4. Acute SAK3 Oral Administration Reduced the Numbers of 4-HNE and Ionized Calcium Binding Adaptor Molecule 1 (IBA-1) Double-Positive Cells in the Hippocampus of OBX Mice

Many reports have declared that microglia are closely related to oxidative stress and excitotoxicity [38]. Therefore, to examine microglial activation level in OBX mice, as well as any effects of acute SAK3 administration on this activation, we measured the number of 4-HNE/Iba-1 double-positive cells, which act as markers of oxidative stress and microglia, respectively, in the hippocampus. In triple immunostaining analyses, the number of 4-HNE/Iba-1 double-positive cells showed a significant increase in the hippocampus of OBX mice (10.67 ± 2.894, *p* = 0.0089 vs. vehicle-treated sham, Figure 4A,C), which was inhibited by acute administration of SAK3 (2.222 ± 0.5212, *p* = 0.0013 vs. vehicle-treated OBX) and donepezil (3.143 ± 0.7377, *p* = 0.0052 vs. vehicle-treated OBX, Figure 4B,C).

## 3. Discussion

In the present study, we observed that SAK3 ameliorated oxidative damage in the OBX mice brains. Moreover, we also discovered that SAK3 could play an important role in restoring cognitive impairments. In this way, we propose that SAK3 acts competitively to rescue AD pathogenesis, benefiting from its efforts to suppress oxidative stress.

The close interaction between oxidative stress and AD pathology was reported in previous research. Moreover, the combination of the amyloidogenic protein Aβ40 to cell membranes was triggered by the oxidation of anionic lipids, thus demonstrating the interior relationship between oxidative stress and Aβ pathogenesis [39]. Actually, some reports demonstrated that Aβ level is significantly increased in OBX mouse brain [8]. In addition, our previous study demonstrated that the enhanced Aβ levels in App^NL-F^ knock-in mouse brain was reduced by a 3-month oral SAK3 administration [40]. Since single and acute administration of SAK3 was carried out in this study, we did not evaluate the Aβ levels in OBX mouse brain. In addition, 4-HNE, a product of lipid peroxidation, altered the conformation of cortical synaptosomal membrane proteins, resulting in the toxicity of the Aβ-enriched brain [41]. Furthermore, the lipoxidation and lower activities of mitochondrial adenosine triphosphate-synthase have been demonstrated in Braak stages I/II, which is defined as the clinically silent stage of AD [42,43]. In addition, some research has revealed that the reduced activation levels of the microRNA miR-124 in response to oxidative stress significantly inhibited the removal of Aβ in microglial cells [44]. It has also been suggested that oxidative stress is immersed in the conformation of neurofibrillary tangles in the very beginning stages of AD [45]. Overall, there is no doubt that oxidative stress coincided with AD pathogenesis. In the present study, an increase in the level of oxidative damage was observed in the brains of AD model mice, which was restored by oral administration of SAK3, demonstrating its positive effects on ameliorating oxidative stress (Figure 2, Figure 3 and Figure 4). Likewise, donepezil also improved the oxidative damage and cognition. However, the improving effects of SAK3 in Y maze task was superior to that of donepezil. In addition, we performed the toxic studies of SAK3 in the GLP levels. There are no obvious toxic effects of SAK3 on the CNS and cardiovascular system by 30 mg/kg by oral administration.

It is believed that neuroinflammation associated with microglial dysfunction is an important characteristic of AD [46]. Microglia display neuroprotective effects when they are inactive in the normal brain but they exhibit neurotoxic effects when shifting into an active type during pathological events, including aging and chemical stimulation [47]. Previous research has revealed that microglia produce oxidation products, such as hydrogen peroxide (H_2_O_2_) and oxidized halides after their activation, thus contributing to oxidative stress in the brain [48]. Indeed, in this study, increased levels of oxidative damage were observed in microglia cells in the hippocampus of AD model mice (Figure 4). Moreover, previous evidence has revealed that M1 microglia, a classical phenotype of activated microglia, produce proinflammatory cytokines including interleukin (IL)-1β, IL-6 and tumor necrosis factor-α upon stimulation with lipopolysaccharides (LPS). These cytokines further promoted the hyperactivation of M1 microglia and thus aggravated inflammation [49]. In contrast, M2 microglia, another microglial phenotype, are considered to exhibit anti-inflammatory effects, as they are involved in the uptake of debris and misfolded proteins, as well as the secretion of anti-inflammatory cytokines, such as IL-4 and IL-10 [26]. Furthermore, accumulated investigations have revealed that α7 nicotinic acetylcholine receptors (α7 nAChRs), a subunit of nAChRs [50] that are closely associated with neuron-related functions such as memory [51], neuron gross and degeneration, play a pivotal role in neuroprotection within microglia [52,53]. More specifically, previous studies demonstrated that stimulation of α7 nAChRs led to a striking suppression of M1 microglia transformation and promoted M2 microglia activation via the Janus kinase 2/signal transducer and activator of transcription 3 signaling pathway, thus exhibiting protective effects under LPS-induced neuroinflammation [49]. Furthermore, we previously observed that oral SAK3 administration potently ameliorated ischemia-induced cognitive impairments and neuronal death by stimulating nAChR-regulated ACh release in the hippocampus of the mouse brain [54]. In addition, activation of nAChRs could ameliorate Aβ-related symptoms through the phosphatidylinositol 3-kinase pathway [55]. Previous reports demonstrated that Aβ-induced reactive oxygen species production was significantly abolished by nAChR stimulation [56]. In the present study, we observed that elevated oxidative stress levels within microglia were potently attenuated by SAK3 treatment (Figure 4). Moreover, memory decline indicated by behavioral tasks in OBX mice was also markedly reversed by SAK3 administration (Figure 2). Based on these studies, we propose that SAK3 triggers the transformation from M1 microglia to M2 microglia by stimulating microglial nAChRs, thus ameliorating oxidative damage and restoring memory dysfunction in AD mouse models. However, in this research, we did not confirm the accurate mechanism about effects of SAK3 on microglial transformation between M1 and M2 types. We will further define the role of SAK3 in transformation from M1 to M2 microglia using biomarkers, respectively, in future.

Taken together, SAK3, a T-type calcium channel enhancer, displayed oxidative stress-suppressing effects in OBX mice. In addition, memory impairment in OBX mice was markedly reduced after acute administration of SAK3. Accordingly, we suggest that the antioxidant effects of SAK3 have the potential to play a prominent role in AD therapy.

## 4. Materials and Methods

### 4.1. Animals and Surgery

Eight-week-old male ddY mice were obtained from Japan SLC, Inc. (Tokyo, Japan), housed together in the same room and given unlimited food and water. The housing environment maintained a temperature of 23 ± 1 °C, humidity of 55 ± 5% and a regular light/dark cycle (light: 9:00–21:00, dark; 21:00–9:00). All experimental procedures using animals were approved by the Committee on Animal Experiments at Tohoku University (2020PhA-7; 17 December 2019). All experiments were carried out under the guidance of the National Institutes of Health Guide for the Care and Use of Laboratory Animals (NIH Publications No. 8023, revised 1978). All reasonable efforts were made to minimize animal stress and the number of animals used. One week later, surgery was performed on the mice, as previously described [34,57]. Fourteen days after surgery, the animals were treated with the reagents described in the following section.

### 4.2. Drug Administration and Experimental Design

SAK3 (Tokyo Chemical Industry Co., Ltd., Tokyo, Japan), the structure of which was shown in Figure 5 and donepezil were dissolved in distilled water at a concentration of 0.5 mg/kg of SAK3 or 1.0 mg/kg separately. The doses of SAK3 and donepezil were selected based on previous research on AD treatment [34,35]. All other reagents were obtained from FUJIFILM Wako Pure Chemical (Osaka, Japan) unless otherwise noted. As shown in Figure 5, all groups of mice were treated with either SAK3 or donepezil at the appropriate dose or distilled water as a vehicle control. The volume of solution administered was 0.1 mL per 10 g of body weight. Thirty minutes after administration, mice were subjected to behavioral tests to measure behavioral impairments, including cognitive deficits and depressive-like behaviors. All mice were subjected to the same behavioral tasks on the same day. The day after finishing their behavioral tasks, the mice were sacrificed and their brains were removed for further study. since we performed the toxic studies of SAK3 in the GLP levels. There are no obvious toxic effects of on the CNS and cardiovascular system by 300 mg/kg by oral administration

### 4.3. Behavioral Tasks

#### 4.3.1. Y-Maze Task

The Y-maze task was used to analyze short-term spatial reference memory conditions in AD model animals [40]. The apparatus consisted of three identical arms (50 × 16 × 32 cm^3^) made of black Plexiglas. Each mouse was placed at the same beginning station at the end of one arm and allowed to move freely in the maze for 8 min. The tool was disinfected by 70% alcohol before every mouse entered the arm. An alternation was defined as consecutive entries into all three arms. The percentage of alternations was calculated as follows [54]:Percentage alterations = actual alternation/(the number of total arm entries − 2) × 100

#### 4.3.2. Novel Object Recognition Task

The novel object recognition task was performed as described previously [40]. In the trial session, each mouse was placed into a test box (35 × 25 × 35 cm^3^), where two objects with the same shape and size were prepared for them to explore for 10 min. In the test session, all of the conditions except that one object were replaced by a novel one 24 h after the trial session for 5 min. Exploration of an object was defined as rearing on, touching or sniffing from a distance of <1 cm from the object. A discrimination index was calculated as the ratio of exploratory contacts to familiar and novel objects.

#### 4.3.3. Step-Through Passive Avoidance Task

The step-through passive avoidance task was performed as described previously [54]. The test box consisted of light (14 × 10 × 25 cm^3^) and dark (25 × 25 × 25 cm^3^) compartments with a stainless-steel rod floor connected to an electronic stimulator (Nihon Kohden, Tokyo, Japan). An adjustable door was set between the two parts. First, each mouse was placed into the light part for 1 min to get used to the environment. Then the door opened, enabling the mouse to enter the dark part upon suffering from electronic stimulation (0.5 mA, 2 s), with the latency time recorded in the trial session. In the test session, which took place 24 h later, mice were placed in the light compartment and step-through latency was recorded until 300 s elapsed to assess retention level.

#### 4.3.4. TST

The TST was performed as previously described [58]. Mice were suspended 50 cm above the floor with their tails stuck to adhesive tapes at a distance of about 1 cm away from the end of the tail. Meanwhile, a timer was used to record the immobile times of mice in 5 min and then 2 min after starting the task (the whole recording was 7 min). The immobile times were recorded under the condition that mice remained completely motionless and hung passively.

#### 4.3.5. FST

The FST was performed as previously described [58]. All groups of mice were placed into glass cylinders separately (height: 20 cm; diameter: 15 cm) containing 25 °C water. Immobility time was recorded for 5 min, 2 min after starting the task. The mouse was considered to be immobile at the time that they stopped struggling and remained floating motionless in the water.

### 4.4. Western Blot Analysis

Immunoblotting analyses were performed as previously described [59,60]. After the behavioral analysis, tissues from the dorsal hippocampus region were dissected and stored at 80 °C before the next experiments were conducted. Frozen samples were homogenized in ice-cold buffer containing 500 mM NaCl, 50 mM Tris-HCl (pH 7.5), 0.5% Triton X-100, 4 mM ethylene glycol-bis(β-aminoethyl ether)-N,N,N′,N′-tetraacetic acid, Ethylenediaminetetraacetic acid 10 mM, 1 mM Na_3_VO_4_, 40 mM Na_2_P_2_O_7_ 10 H_2_O, 50 mM NaF, 100 nM calyculin A, 50 μg/mL leupeptin, 25 μg/mL pepstatin A, 50 μg/mL trypsin inhibitor and 1 mM dithiothreitol. Samples were then centrifuged at 15,000 rpm for 10 min at 4 °C to remove insoluble material. Protein concentration was determined using Bradford’s assay and samples were boiled for 3 min at 60 °C with Laemmli’s sample buffer without 2-mercaptoethanol (0.38 M Tris-HCl, pH 6.8, 15% glycerol, 12% SDS and 0.05% bromophenol blue). Equivalent amounts of protein were loaded onto SDS-polyacrylamide gels and transferred to Immobilon polyvinylidene difluoride membranes. After blocking with T-TBS solution (50 mM Tris-HCl, pH 7.5, 150 mM NaCl and 0.1% Tween 20) containing 5% skim milk powder for 0.5 h at room temperature, membranes were incubated with anti-4-hydroxy-2-nonenal (4-HNE) monoclonal antibody (1:500; MHN-100P, clone HNE-J2, JaICA, Shizuoka, Japan) or anti-nitrotyrosine monoclonal antibody (1:1000; 05233, clone 1 A6, Sigma-Aldrich, St. Louis, MO, USA) or anti-β-actin monoclonal antibody (1:5000; A5441, clone AC-15, Sigma-Aldrich) for a day at 4 °C. A T-TBS solution was used to dilute all antibodies. The next day, membranes were incubated with the appropriate horseradish peroxidase-conjugated secondary antibodies (1:5000; Southern Biotech, Birmingham, AL, USA) diluted in T-TBS solution for 2 h at room temperature after being washed with a T-TBS solution. The membranes were developed using an enhanced chemiluminescence immunoblotting detection system and visualized on a radiograph (Fuji Film, Tokyo, Japan). Protein expression levels were quantified using Image Gauge version 3.41 (Fuji Film).

### 4.5. Immunofluorescence Staining

Immunofluorescence staining was performed as previously described [60]. For immunohistochemical analyses, mice were perfused with ice-cold phosphate-buffered saline (PBS, pH 7.4) and 4% paraformaldehyde (PFA) under anesthesia. Brains were removed and fixed in 4% PFA overnight at 4 °C. Next, we used a vibratome to obtain brain coronal sections of approximately 50 μm in thickness (DTK-1000, Dosaka EM Co. Ltd., Kyoto, Japan). Hippocampal sections 1.7–2.2 mm away from bregma and seven sections per mouse were used for the immunofluorescence analyses. After washing with PBS for 20 min, brain sections were permeabilized with 0.1% Triton X-100 in PBS. Brain sections were then immersed in 30% H_2_O_2_ diluted with PBS to remove endogenous peroxidase. Next, sections were blocked with PBS containing 1% bovine serum albumin and 0.3% Triton X-100 for 1 h at room temperature after washing with PBS. After blocking, the sections were incubated with primary antibodies for 3 days at 4 °C. These consisted of mouse anti-4-HNE (1:500; JaICA), mouse anti-nitrotyrosine (1:1000), rabbit anti-NeuN (1:500; Millipore, Billerica, MA, USA) and mouse anti-Iba1 (1:2000; 019–19741, Wako) diluted in a blocking solution. After washing with PBS, sections were incubated with secondary antibodies (Alexa 488 anti-rabbit IgG and Alexa 594 anti-mouse IgG; 1:500 in blocking solution; Invitrogen, Waltham, MA, USA) away from light overnight at 4 °C. Subsequently, sections were washed with PBS and then permeabilized with 4,6-diamidino-2-phenylindole, usually called by DAPI (Thermo Fisher Scientific, Waltham, MA, USA), diluted in PBS (1:1000), for 5 min under the exclusion of light at room temperature. After several washes with PBS, sections were mounted in Vectashield (Vector Laboratories, Inc., Burlingame, CA, USA) and immunofluorescence images were obtained using a confocal laser scanning microscope (TCS SP8, Leica Microsystems, Wetzlar, Germany). As shown in Figure 3, we calculated the numbers of 4-HNE or nitrotyrosine positive cells respectively. As shown in Figure 4, the number of 4-HNE/Iba-1 double positive cells were evaluated as well. All the Immunofluorescence images were quantitatively analyzed using Image J software.

### 4.6. Statistical Analysis

All data are presented as mean ± standard error of mean (SEM). The comparison between two groups was conducted using an unpaired Student’s *t*-test. Comparisons among multiple groups were evaluated by one-way or two-way analysis of variance followed by Tukey’s post-hoc test using GraphPad Prism 7 (GraphPad Software, Inc., La Jolla, CA, USA).

## Figures and Tables

**Figure 1 ijms-22-00741-f001:**
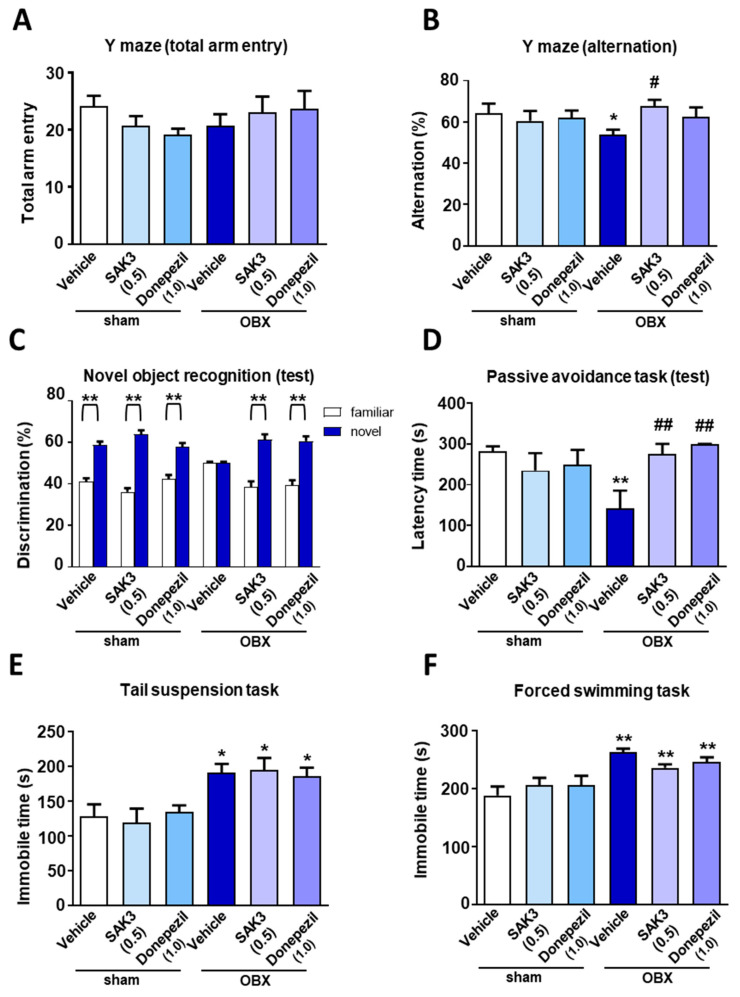
The effects of acute SAK3 and donepezil administration on spatial memory, cognitive functions and depressive-like behaviors in OBX mice: (**A**) Number of total arm entries and (**B**) alternations in a Y-maze task (n = 10 per group). Discrimination index of object exploration during (**C**) the test session in a novel object recognition task (n = 10 per group). Latency times before entering into the dark compartment from the light compartment during (**D**) the test session in a passive avoidance task (n = 10 per group). The reduction in immobility time between sham and OBX mice are presented in the TST (**E**) and FST (**F**) but no difference was observed between sham and OBX mice after acute SAK3 and donepezil administration (n = 10 per group). Error bars represent standard error of mean. * *p* < 0.05 vs. vehicle-treated sham mice; # *p* < 0.05 vs. vehicle-treated OBX mice, ** *p* < 0.01 vs. the familiar group by Student’s *t*-test; ** *p* < 0.01 vs. vehicle-treated shame mice; ## *p* < 0.01 vs. vehicle-treated sham mice.

**Figure 2 ijms-22-00741-f002:**
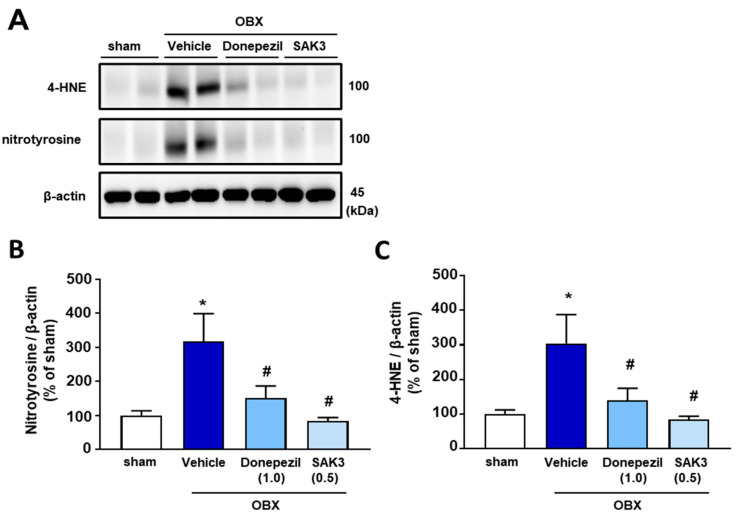
Elevated oxidative stress in OBX mice is suppressed by SAK3 and donepezil treatment in immunoblotting: (**A**) Representative image of western blots probed with antibodies against 4-HNE, nitrotyrosine or β-actin in the hippocampus; (**B**,**C**) Quantitative analyses of 4-HNE or nitrotyrosine protein levels (n = 10 per group). Error bars represent standard error of mean. * *p* < 0.05 vs. vehicle-treated sham mice; # *p* < 0.05 vs. vehicle-treated OBX mice.

**Figure 3 ijms-22-00741-f003:**
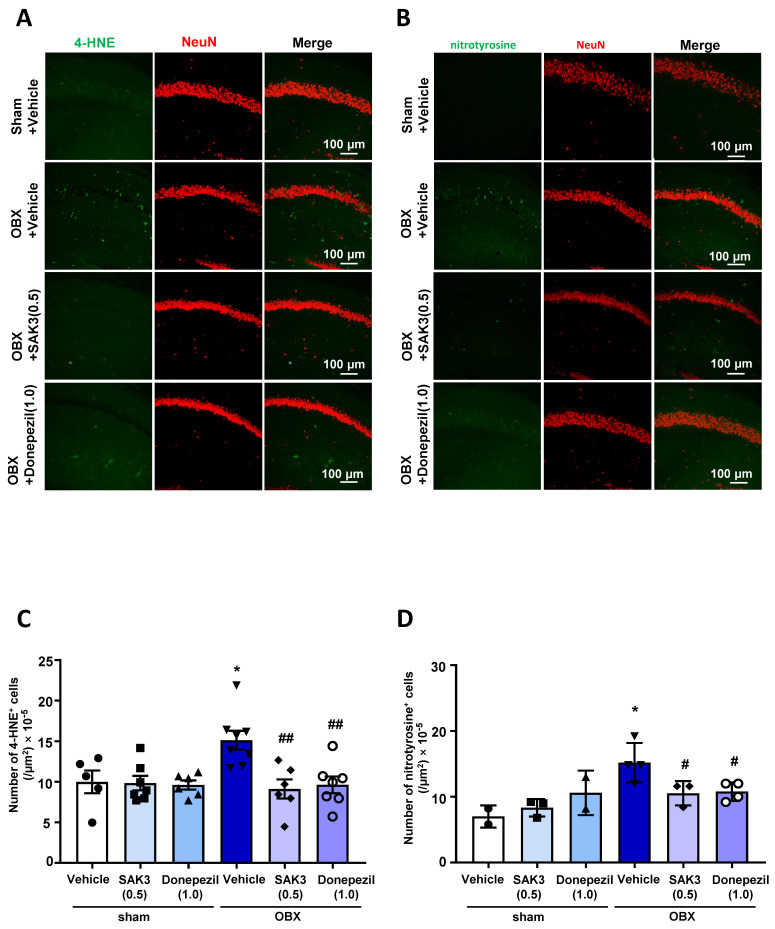
Immunohistochemistry reveals that SAK3 and donepezil inhibited OBX surgery-induced oxidative damage: Representative images of fluorescence immunostaining with anti-4-HNE antibody (**A**) or anti-nitrotyrosine antibody (**B**) in the hippocampal CA1 regions. Scale bars: 100 μm; Quantitative analyses of the number of 4-HNE-positive cells (**C**) or nitrotyrosine-positive cells (**D**) in the CA1 (n = 5–8 per group). Error bars represent standard error of mean. The scattered solid circles, squares, regular triangles, inverted triangles, rhombuses and hollow circles inside the histograms were plotted using the real values to show the results clearly. * *p* < 0.05 vs. vehicle-treated sham mice; ## *p* < 0.01 vs. vehicle-treated OBX mice; # *p* < 0.05 vs. vehicle-treated OBX mice.

**Figure 4 ijms-22-00741-f004:**
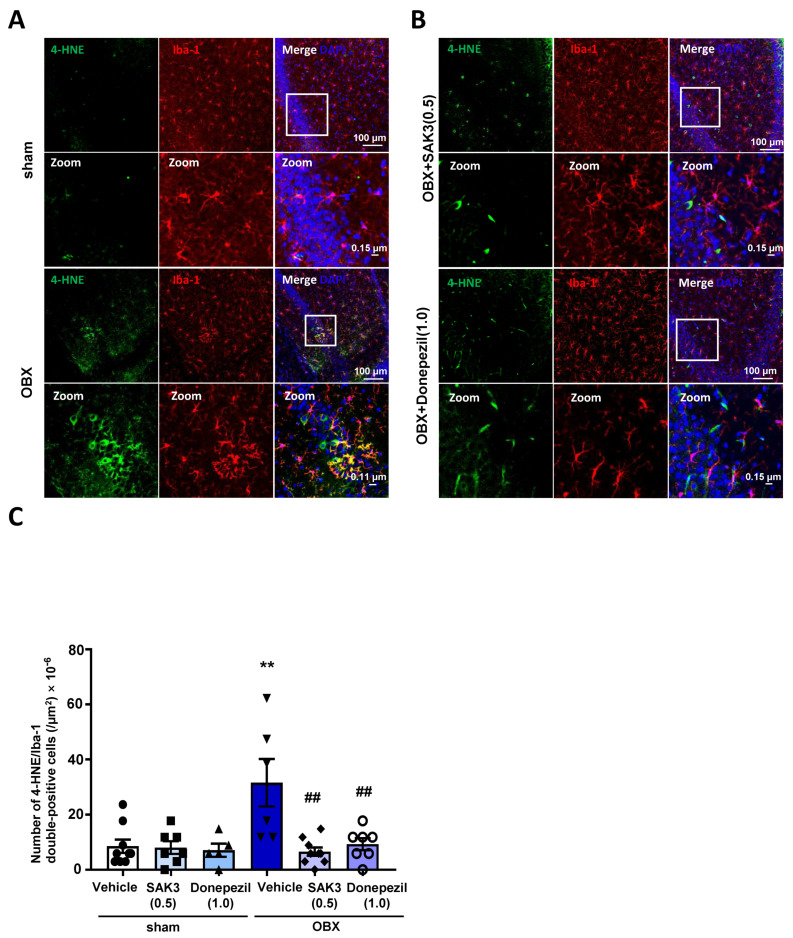
Increased microglial activation levels were restored by acute oral administration of SAK3 and donepezil in the hippocampus, as revealed by immunohistochemistry: (**A**,**B**) Representative images of triple fluorescence immunostaining with anti-4-HNE, anti-Iba-1 antibody and DAPI in the hippocampus. Scale bars: 100 μm, 0.15 μm or 0.11 μm; (**C**) Quantitative analyses of the number of 4-HNE/Iba-1 positive cells in the hippocampus (n = 5–8 per group). Error bars represent standard error of SEM. The scattered solid circles, squares, regular triangles, inverted triangles, rhombuses and hollow circles inside the histograms were plotted using the real values to show the results clearly. ** *p* < 0.01 vs. vehicle-treated sham mice; ## *p* < 0.01 vs. vehicle-treated OBX mice.

**Figure 5 ijms-22-00741-f005:**
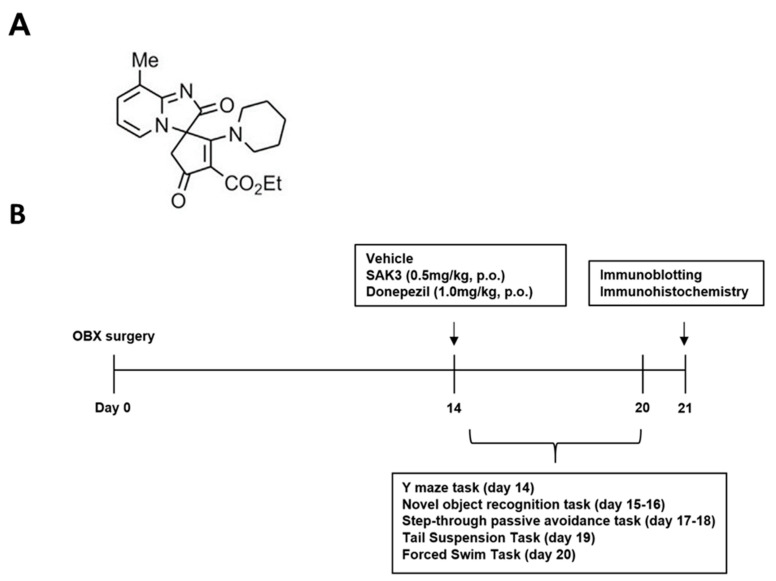
Experimental protocol for this study: (**A**) chemical structure of SAK3; (**B**) Experimental schedule for the present study. From day 14, 8-month-old ddY sham and OBX mice were subjected to a behavioral task 30 min after vehicle, SAK3 (0.5 mg/kg, p.o.) or Donepezil (1.0 mg/kg, p.o.) administration. After finishing the behavioral task, mice were sacrificed and their brains were removed for use in various experiments.

## Data Availability

The data that support the findings of this study are available from the corresponding author upon reasonable request.

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
