# Peer review of "Single Administration of the T-Type Calcium Channel Enhancer SAK3 Reduces Oxidative Stress and Improves Cognition in Olfactory Bulbectomized Mice"

_ijms, 2021, doi:10.3390/ijms22020741_

Round 1
Reviewer 1 Report
The authors describe their studies which aimed to investigate the protective role of ethyl-8-methyl-2,4dioxo-2-(piperidin-1-yl)-2H-spiro[cyclopentane-1,3-imidazo[1,2-a]pyridin]-2-ene-3-carboxylate (SAK3) in olfactory bulbectomized mice (OBX mice) as a model of Alzheimer’s disease (AD). The present study demonstrated that SAK3 decreased 4-hydroxy2-nonenal and nitrotyrosine protein levels in the hippocampus of OBX mice, similar to the effects of donepezil. In the same way, markers for oxidative stress in microglia were reduced with SAK3, and as a consequence, they observed an improvement in spatial memory and cognitive functions. Based on these data, they proposed SAK2 as a potential therapeutic agent against AD.
The manuscript is well written, and comprehensive data have been analyzed and presented. It is acceptable for publication in the International Journal of Molecular Sciences with major revision:
1. It Is important to add more data about in vitro results with this molecule in the introduction.
2. Why the authors did not measure the reactivity of NeuN? It will be really interesting, and it will add more value to the results, considering that it has been reported cell death in hippocampus areas in Alzheimer’s disease models. Besides, why the authors don’t evaluate AD markers as beta-amyloid?
3. Considering the explanation of Yamazaki et al., 2012 about the DDY mice: “The ddY strain differs from the DDY strain because the DDY strain has been established as an inbred strain from the ddY colony at Yoken (note that the inbred strain DDY is spelled with all uppercase letters and with lowercase letters in the name of outbred strain ddY).” Why the authors choose DDY mice in this study if they reported the ddY mice in its supportive previous study?
4. In the methods, they should include the range of bregmas that was used for the study and describe how they did the quantification for the immunofluorescence images
5. Catalog numbers from reagents and antibodies should be mentioned in the methods for all the substances. Some of them are missing:4-HNE
6. Statistics should be improved. Normality tests were done for the data?
7. This is a very interesting manuscript, in which in convincing way protective properties of SAK3, pointing the usefulness of its application in practice. However, I consider that the negative role of SAK3 should be mentioned in the discussion.
8. More detail about how the behavioral tests were done will be useful. The rationale and selection process for the order and sequence, assuming the same animals are run through both assays. Keeping in mind, that has been reported that exposure to a battery of behavioral test and extensive handling could allow serious discomfort and stress. Therefore, the data obtained could be affected. Consider elaborate a discussion point about it.
9. In the discussion, the author mentions, “we failed to distinguish SAK3 from donepezil in suppressing oxidative damage in OBX mice brains” however they don’t delve into this point in the discussion.
10. I consider the authors don’t have enough markers to confirm “that elevated oxidative stress within M1 microglia (indicated by Iba-1 expression) was attenuated by SAK3 administration” M1 and M2 phenotypes have been extensively discussed in the literature, so I suggest more careful to ensure these phenotypes.
Author Response
Response to Reviewer 1
- It Is important to add more data about in vitro results with this molecule in the introduction.
Ans: According to the comment, we added the in vitro findings in lines 83-88, as followed. “Based on whole cell patch-clamp analysis, SAK3 (0.01-10 nM) significantly enhanced Cav3.1 current in neuro2A cells ectopically expressing Cav3.1. SAK3 (0.1-10 nM) also enhanced Cav3.3 but not Cav3.2 currents in the transfected cells [34]”. Besides, the chemical structure of SAK3 was shown in Fig. 5(a).
- Why the authors did not measure the reactivity of NeuN? It will be really interesting, and it will add more value to the results, considering that it has been reported cell death in hippocampus areas in Alzheimer’s disease models. Besides, why the authors don’t evaluate AD markers as beta-amyloid?
Ans: We agree with the comment. There is a report that the cell number in the hippocampal CA1 regions is decreased 4 weeks after OBX rats (Yurttas C et al., 2017 Brain Res Bulletin). However, as shown in Fig. 3, we did not observe the significant changes in NeuN positive cell numbers at 3 weeks after OBX in mice. Likewise, our previous study using OBX mice revealed the no significant changes in the neuronal marker levels such as CaMKII, AMAP receptor and synapsin I in OBX mouse hippocampal CA1. However, the NeuN reactivity should be evaluated in mice in future study.
Regarding Ab levels, some reports demonstrated that there is a significantly increased level of Aβ in OBX mouse brain (Aleksandrova, I.Y. et al., Increased level of beta-amyloid in the brain of bulbectomized mice. Biochemistry 2004, 69, 176–180). We previously documented that Aβ levels in AppNL-F knock-in mice are significantly reduced by oral SAK3 administration for 3 months (Izumi, H. et al. Neuroscience 2018, 377, 87–97). Because the present study was carried out by acute and single administration of SAK3, we did not evaluate the level of Aβ in OBX mouse brain. We explain the SAK3 effects on the Aβ production in App knock-in mice (lines 174-178).
- Considering the explanation of Yamazaki et al., 2012 about the DDY mice: “The ddY strain differs from the DDY strain because the DDY strain has been established as an inbred strain from the ddY colony at Yoken (note that the inbred strain DDY is spelled with all uppercase letters and with lowercase letters in the name of outbred strain ddY).” Why the authors choose DDY mice in this study if they reported the ddY mice in its supportive previous study?
Ans: We corrected to ddY and sorry for simple our mistake.
- In the methods, they should include the range of bregmas that was used for the study and describe how they did the quantification for the immunofluorescence images
Ans: According to the comment, we described that hippocampal sections 1.7-2.2 mm away from bregma and seven sections per mouse were used for this study. After consecutive operations, as shown in Fig. 3, we calculated the numbers of 4-HNE or nitrotyrosine positive cells respectively. As shown in Fig. 4, the number of 4-HNE/Iba-1 double positive cells were evaluated as well. Besides, we have added them into the Methods (lines 316-317 and 332-333).
- Catalog numbers from reagents and antibodies should be mentioned in the methods for all the substances. Some of them are missing:4-HNE
Ans: According to the comments, we added the catalog number of 4-HNE and other antibodies (lines 302-304).
- Statistics should be improved. Normality tests were done for the data?
Ans: we agree with the comment. In Fig2, we normalized the levels of nitortyrosine and 4-HNE by levels of b-actin. We also normalized the number of positive cells by the size of area after counting cells in Figs, 3 and 4.
- This is a very interesting manuscript, in which in convincing way protective properties of SAK3, pointing the usefulness of its application in practice. However, I consider that the negative role of SAK3 should be mentioned in the discussion.
Ans: Thank you for the encouragement. Since we performed the toxic studies of SAK3 in the GLP levels. There are no obvious toxic effects of SAK3 on the CNS and cardiovascular system by 30mg/kg by oral administration and this is described in the text (lines 190-191).
- More detail about how the behavioral tests were done will be useful. The rationale and selection process for the order and sequence, assuming the same animals are run through both assays. Keeping in mind, that has been reported that exposure to a battery of behavioral test and extensive handling could allow serious discomfort and stress. Therefore, the data obtained could be affected. Consider elaborate a discussion point about it.
Ans: According to the comments, we adopted the order of behavioral tests. We first performed Y-maze task and Novel object recognition task to reduce stress to mice. Then, we carried out TST and FST in the order.
- In the discussion, the author mentions, “we failed to distinguish SAK3 from donepezil in suppressing oxidative damage in OBX mice brains” however they don’t delve into this point in the discussion.
Ans: According to the comment, we added the discussion in the text (lines 188-191).
- I consider the authors don’t have enough markers to confirm “that elevated oxidative stress within M1 microglia (indicated by Iba-1 expression) was attenuated by SAK3 administration” M1 and M2 phenotypes have been extensively discussed in the literature, so I suggest more careful to ensure these phenotypes.
Ans: W agree with the comment. We did not define M1 and M2 phenotypes by SAK3 treatment in the present study. This is very important to address the SAK3 mechanism. We will define the M1 and M2 phonotype in the next paper.
Reviewer 2 Report
The study by Yuan D et al titled “Single administration of the T-type calcium channel enhancer SAK3 reduces oxidative stress and improves cognition in olfactory bulbectomized mice” is based on previous studies. Now their study is focused in investigate the antioxidative effects of SAK3 in olfactory bulbectomized mice (OBX) mice, as a human Alzheimer Disease (AD) model.
They investigated the antioxidative effects of SAK3 in olfactory bulbectomized mice (OBX mice) and explored the interaction between oxidative stress and AD pathogenesis.
Their main novel results obtained have been that increased protein levels of oxidative stress markers in the microglial cells were rescued by SAK3 in the same way as donepezil.
The Discussion is clear and fully explained. However, I suggest that the authors review the manuscript.
Main comments
- In several figures, the authors mentioned “veh” referred to vehicle-treated OBX mice or vehicle-treated OBX mice. They should change these terms: “veh” by full word “vehicle” and “Done.” by “Donepezil”
- In Fig 4 the authors should add a scale bar in the Zoom images.
- The authors have suggested that the antioxidative effects of SAK3 and donepezil are among the neuroprotective mechanisms in AD pathogenesis. Have the authors analyzed the survival or protection in the neuronal population in OBX mice? They could use a death marker (i.e. TUNEL or cleaved Caspase-3) under different conditions. They should provide these results.
- Eight-week-old male DDY mice were used in OBX-model. The authors should discuss the possible effects of SAK3 and donepezil in older mice (i.e 4 months-old)? Do the authors know the effect in female mice?
Author Response
Response to Reviewer 2
- In several figures, the authors mentioned “veh” referred to vehicle-treated sham mice or vehicle-treated OBX mice. They should change these terms: “veh” by full word “vehicle” and “Done.” by “Donepezil”
Ans: According to the comment, we change to the full words in all the figures.
- In Fig 4 the authors should add a scale bar in the Zoom images.
Ans: We added the scale bar in the zoom images were already made up in Fig. 4 at the page of 7.
- The authors have suggested that the antioxidative effects of SAK3 and donepezil are among the neuroprotective mechanisms in AD pathogenesis. Have the authors analyzed the survival or protection in the neuronal population in OBX mice? They could use a death marker (i.e. TUNEL or cleaved Caspase-3) under different conditions. They should provide these results.
Ans: We agree with the comment. Previous investigations demonstrated that apoptotic and TUNEL positive cells were increased 24 hours after olfactory bulbectomy to the rats. However, the elevated apoptosis levels came back to normal levels 36 hours after olfactory bulbectomy [Pope, K. Olfactory system modulation of hippocampal cell death. Neurosci. Lett. 2007, 422, 13–17]. In the present study, as shown in Fig. 3, there is no significant change in numbers of NeuN-positive cells. Since our results from acute effects, we did not evaluate the apoptosis levels in this study.
- Eight-week-old male DDY mice were used in OBX-model. The authors should discuss the possible effects of SAK3 and donepezil in older mice (i.e 4 months-old)? Do the authors know the effect in female mice?
Ans: Previous out study showed that spine abnormalities and memory impairment were improved by chronic oral SAK3 administration in 9 month old female mice in AppNL-G-F/NL-G-F knock-in mice [Izumi, H.et al., IJMS, 2020, 21, 3833; Neuroscience 2018, 377, 87–97.]. In this context, SAK3 improves cognition in AD female mice.
Round 2
Reviewer 1 Report
The authors addressed all the comments and suggestions in a good way.